# Current Research Trends, Hotspots, and Frontiers of Physical Activity during Pregnancy: A Bibliometric Analysis

**DOI:** 10.3390/ijerph192114516

**Published:** 2022-11-04

**Authors:** Yanbing Zhou, Xian Guo, Jinhao Mu, Jingying Liu, Hongying Yang, Chenxi Cai

**Affiliations:** 1School of Art, Beijing Sport University, Beijing 100084, China; 2Sport Science School, Beijing Sport University, Beijing 100084, China; 3Library of Beijing Sport University, Beijing Sport University, Beijing 100084, China; 4State Key Laboratory of Molecular Vaccinology and Molecular Diagnostics, School of Public Health, Xiamen University, Xiamen 361005, China

**Keywords:** physical activity, pregnancy, bibliometric analysis

## Abstract

Purpose: Physical activity (PA) during pregnancy has been proven beneficial to pregnant women, with a significant effect on ameliorating many severe gestational complications. This work aimed to reveal current research trends, hotspots, and future frontiers in PA during pregnancy. Methods: Software CiteSpace was used to perform a bibliometric analysis with 1415 publications in the Web of Science Core Collection. Results: the number of published papers on PA during pregnancy has increased gradually by year. The United States has made the most significant contribution to the research on this topic, ranking first in the world in both the number and centrality of research. A total of 54 articles (3.8%) were published in 2022. A majority of publications were research articles (n = 1176, 78.9%). The authors and institutions of the research published have more inter-country collaborations. Different patterns of PA, prevention, and amelioration of pregnancy complications are major research hotspots. Improvement of sedentary behaviour, lifestyle intervention through leisure-time PA, and preterm care are major research frontiers and have received extensive attention in recent years. Conclusions: The current scientometric study presents an overview of PA during pregnancy research conducted throughout the preceding decades. The conclusions of this work might serve as a reference for researchers who are interested in this field.

## 1. Introduction

Pregnancy is a special life experience for most women. Increased blood volume and heart rate, weight growth, and also a transition in the center of gravity are all typical hormonal and physiological changes that can possibly occur. Many severe complications may be induced such as gestational diabetes mellitus (GDM), pre-eclampsia, and hypertension [1,2,3].

Physical activity (PA) during pregnancy was already noticed and considered beneficial during the 17th and 18th centuries [4]. However, whether it should be promoted or not remained controversial. Pregnant women were encouraged to exercise and walk to increase the volume of PA, but many types of physical activities like dancing and horse-riding were prohibited at the same time because it was believed that it might lead to a fetus’ head striking a mother’s pelvis [4]. One of the earliest PA guidelines for pregnant women was issued in 1912, stating walking was the best kind of exercise [5]. Many of the first studies that focused on the relationships between birth weight and PA were published in the 1980s, ascribing higher levels of occupational and household PA to lower birth weights [6]. Many epidemiological studies have assessed the relationship between PA and pregnancy outcomes during the last few decades [6,7], but effective evidence for adverse pregnancy outcomes is still lacking [7]. In 1985, the American College of Obstetricians and Gynecologists (ACOG) published the first guidelines on prenatal PA, emphasizing the safety of aerobic exercise, but recommending against high-intensity PA, such as running [8]. The revision by ACOG in 1994 removed the upper limit of the heart rate and duration of exercise [9]. In 2002, ACOG recommended pregnant women with no complications to commit 30-min PA with moderate intensity daily [10]. Similarly, the U.S. Department of Health and Human Services (USDHHS) issued the US PA Guidelines in 2008, suggesting pregnant women without medical complications perform at least 150 min of moderate-intensity PA per week [11].

PA has been globally recommended for decades, yet few populations achieve the suggested volume because of complicated reasons including but not limited to intrapersonal barriers (including pregnancy symptoms, family and child-rearing responsibilities, lack of personal motivation, etc.), additional interpersonal barriers (including lack of social support, overprotective family members, etc.) and environmental, organizational, and political factors [12]. Therefore, we utilized CiteSpace software to implement a bibliometric analysis in research of PA during pregnancy to reveal current research trends, hotspots, and future frontiers in this field.

Bibliometric analysis is a widely used research method that investigates particular structures, characteristics, and principles of science and technology by examining the research history, turning points, and most noteworthy research trends [13]. CiteSpace is a flexible, powerful, and widely used software in bibliometric analysis that uses a mapping knowledge domain to provide a vast amount of information about important research power, hotspots, and global trends [14,15]. After going through a systematic literature search, no bibliometric analysis studies on PA during pregnancy have been found published. This bibliometric paper analyzes the research status of PA during pregnancy through the CiteSpace visualization operation, so as to provide a theoretical basis for the promotion of PA during pregnancy.

## 2. Materials and Methods

### 2.1. Data Source

The search strategy demonstrated in Figure 1 was performed in the advanced search of Web of Science Core Collection of the Web of Science on 4 July 2022. In order to retrieve more precise and relevant results, wildcard characters (question marks) were used along with positional operators (double quotation marks, etc.) and logical operators (brackets, ‘AND’, ‘OR’, etc.). A total of 1415 research articles and reviews published in all languages from the inception of the database to 2022 were retrieved and extracted with full records and cited references. 

### 2.2. Research Method

The CiteSpace 6.1.R2 software was utilized for generating knowledge maps and analyzing information involving country, institution, cited author, cited journal, keyword, and burst word. The detailed settings on CiteSpace software were as follows: time slicing (From 2010 January to 2022 April), years per slide (1), links (strength: cosine; scope: within slices), selection criteria (g-index: k = 25; select top 50 levels of each slice; select top 10% of each slice; the maximum number of selected items per slide: 100), pruning (pathfinder and pruning slices network).

## 3. Results

### 3.1. Temporal Distribution of Literature

The initial search for PA during pregnancy resulted in 1415 research articles and reviews published in all languages from the inception of the database to 2022. A total of 54 articles (3.8%) were published in 2022. A majority of publications were research articles (n = 1176, 78.9%). According to the publication years, the number of published articles and reviews on PA during pregnancy increased significantly over the studied period with some fluctuations (Figure 2). The list of the top ten categories of research is shown in Figure 3, and the most important research subject is Obstetrics gynecology.

### 3.2. Country and Institution Distribution of Documents

Country and institution distribution were shown in Figure 4 and Figure 5 respectively. Table 1 summarized the top ten productive countries and institutions in research on PA during pregnancy. The size of nodes represents the number of publications of different countries and institutions. The shorter and coarser line between two nodes suggests closer cooperation between countries and institutions. The colors of lines and nodes represent the time slice directly with colder colors indicating the latest years. The USA (402, 35.5%) ranked first in the number of publications, followed by Canada (131, 11.6%) and Australia (113, 10.0%). As for institutions, the University of Alberta (30, 14.6%) ranked in first place, followed by the University of Carolina (29, 14.1%) and the University of Western Ontario (21, 10.2%). Furthermore, there was internationally extensive cooperation across countries and institutions.

### 3.3. Author and Journal Distribution of Documents

Figure 6 is the co-occurrence map of authors in research on PA during pregnancy. The size of nodes represents the number of publications of different authors. The shorter and coarser line between two nodes suggests the most collaboration between individual authors. The colors of lines and nodes represent the time slice directly with the colder color indicating the latest years. The list of top ten authors and journals is shown in Table 2. The most active author is Barakat Ruben from USA Northwell Health’s Cancer Institute, who was found to have contributed to 28 papers in this field. The following productive authors are Margie Davenport (21) from Canada’s University of Alberta and Kristi Adamo (17) from Canada’s University of Ottawa. The periodical with the largest number of articles published is the American Journal of Obstetrics and Gynecology (637, 14.8%), followed by Obstetrics and Gynecology (631, 14.7%) and Medicine and Science in Sports and Exercise (509, 11.9%). The influence factors of journals are between 3.007–79.321, which indicates relatively high quality and great influence.

### 3.4. Current Research Trends and Hotspots in Research of PA during Pregnancy

The co-occurrence map of keywords was generated as in Figure 7. The size of nodes represents the number of publications on different topics. The shorter and coarser line between two nodes suggests a closer connection between different topics. The colors of lines and nodes represent the time slice directly with the colder color indicating the latest years. The top ten most cited keywords are as follows: PA (365), risk (321), pregnancy (338), exercise (151), aerobic exercise (126), health (118), prevalence (96), gestational diabete millitus (95), preterm birth (72), and association (70).

The largest nine clusters based on the co-occurrence map of keywords were obtained in the form of cluster view (Figure 8) and timeline view (Figure 9) respectively. The timeline view map demonstrates time points of appearance and corresponding duration time of each cluster with nodes and arcs on a horizontal time axis. The size of nodes represents the number of publications while the length of the arcs reflects the duration time.

### 3.5. Frontiers in Research of PA during Pregnancy

The top twenty keywords with the strongest citation bursts from 2010 to 2022 were identified in Table 3. The blue part symbolizes the time interval; meanwhile, the red part represents the duration of the citation burst, thus demonstrating the shift of research hotspots over time. Noticeably, the burst of activity pattern lasted the longest for five years; in the meantime, weight, sedentary behavior, and systematic review lasted for four years individually. Sedentary behavior, systematic review, symptom, lifestyle, and prenatal care are currently the latest burst words begun in 2018, 2019, and 2020 to 2022. 

## 4. Discussion

According to the operation results of CiteSpace, the number of published papers on the topic of PA during pregnancy has shown a significant upward trend, from 69 in 2010 to 181 in 2021. The most published categories are obstetrics gynecology, public environmental occupational health, and sport sciences, with strong integration of academic disciplines. The United States is the most prolific country on this topic, with 402 published studies accounting for 35.5% of the total studies. University of Alberta and University of Carolina are globally the most productive institutions on this topic, comprising 30 and 29 publications, respectively, for 14.6% and 14.1% of the total. Professor Barakat Ruben, affiliated with Northwell Health’s Cancer Institute, has published a total of 28 studies on this topic, making him the most prolific researcher. There is a total of 637 (14.8%) related studies published in the American Journal of Obstetrics and Gynecology, which published the most research on this topic, with an impact factor of 8.661. Risk (321) and exercise (151) are the most cited keywords besides physical activity (365) and pregnancy (338). #Gestational diabetes is the largest cluster besides #physical activity. 

The development and evolution of research hotspots can be obtained by sorting out the clusters of keywords and burst keywords from CiteSpace operation. The aforementioned clusters can be generally divided into three aspects including PA patterns (#0 physical activity, #8 muscle strengthening), lifestyle, risk factors (#3 sitting time, #4 preterm birth, #6 risk factors), and pregnancy complications (#1 gestational diabetes, #2 oxidative stress, #5 insulin sensitivity). The noticeable keywords with the strongest burst in 2010–2015 included activity pattern, exercise, maternal exercise, endurance exercise, and leisure-time PA, which indicated that activity pattern was the prior research focus. In 2016–2022, the burst keywords of GDM and maternal obesity demonstrated that the research focus during this period had gradually shifted to the improvement of pregnant complications. Moreover, keywords such as lifestyle intervention, sedentary behavior, lifestyle, and prenatal care, with the strongest burst, entailed that the effect of lifestyle on pregnancy was also a research hotspot. 

The amelioration of pregnancy complications by PA has been another research hotspot in recent years. “Risk factor” is a high-frequency keyword, with 321 citations and one main cluster, which mainly addressed GDM, gestational weight gain, anxiety, and depression, lower back pain, pelvic girdle pain, fetal responses, and birth weight. GDM has emerged as a highly cited keyword, a strong burst keyword, and a main cluster, showing it as a primary research hotspot. While weight, maternal obesity, and weight loss are all strong burst keywords, as well indicating that gestational weight gain and weight control are both essential research spots. Globally, half of all the females who are at childbearing age are overweight or obese [16], which potentially will trigger excessive gestational weight gain and increases the risk of developing GDM [16,17,18]. GDM is related to a higher incidence of negative pregnancy outcomes and a long-term risk of childhood obesity and type-2 diabetes in both mother and the infant. Nevertheless, excessive gestational weight gain, GDM, as well as the possible complications of obesity during pregnancy could be minimized with PA [18]. The 2018 PA Guidelines Advisory Committee reported a significant inverse relationship between PA and weight gain, risk of gestational diabetes mellitus or preeclampsia, and symptoms of depression or anxiety [13]. A former study suggested that for pregnant women with GDM, any form of PA that is sufficiently intense and prolonged can be beneficial [19]. A meta-analysis demonstrated that participating in prenatal PA reduces the risk of having GDM by 40% [20]. Anxiety and depression can be commonly observed in pregnant women, which may have a negative impact on the health of both the mother and fetus [21,22]. An incidence of 16% of depressive symptoms, 5% of severe depressive symptoms, and 27% of lifted prenatal anxiety was demonstrated in previous studies [22]. PA has been proven able to play a role as a psychotherapeutic to attenuate depression and anxiety by changing neurotransmitter and hormone levels that are linked to depression and enhancing the encouragement of self-efficacy [23,24,25,26]. Lower back pain and pelvic pain happened frequently in two-thirds of pregnant women as pregnancy advances [27]. A meta-analysis illustrated that multiple types of exercise obtain functions ameliorating back pain and pelvic pain involving aerobic exercise, muscle strengthening exercise, flexibility exercise, and stretching exercise [28]. Fetal responses and birth weight have been addressed a lot with the citations of health and preterm birth. On one hand, researchers have illustrated that fetuses give response during or after exercise as light-to-moderate increases in fetal heart rate of 10–30 beats per minute over the baseline [29,30]. On the other hand, despite this, women who tend to exercise vigorously during the third trimester have a greater possibility to deliver infants with weights 200–400 g less than those who do not [31,32]. Yet neither has observed an increased risk of fetal growth or preterm birth; rigorous exercise has been proven safe for fetuses and mothers in the second trimester, whether they are physically active or not [29,30,31,32]. 

“PA,” “exercise,” and “aerobic” are all highly cited keywords with citations of 365, 151 and 126, whilst “activity pattern,” “exercise, maternal exercise,” and “endurance exercise” are all strong burst keywords. Moreover, #0 physical activity and #8 muscle strengthening are both prevalent main clusters. Aerobic exercise was used the most widely during pregnancy [33]. Regular aerobic exercise during pregnancy has been proven efficient to improve or maintain physical fitness. Specifically, simple activities such as walking, cycling, swimming, or other modified activities like yoga, are all encouraged for pregnant women to regularly perform in every trimester [34,35]. Current recommendations state that healthy pregnant women should engage in 150 min or more of moderate-intensity aerobic activity each week [36,37]. This exercise should be carried out over the duration of the week and modified as necessary for health. For instance, pregnant women who engaged in aerobic exercise for 30–60 min per day, 2–7 times per week, were observed having a significantly lower risk of prenatal hypertensive disorders, gestational hypertension, and cesarean delivery than the sedentary group [38]. 

Guidelines reported that healthy pregnant women who are very active, or frequently engage in vigorous-intensity aerobic activity can continue the intensive training [33,36], such as strength training, or high-intensity interval training (HIIT). Research indicated that resistance exercise can achieve perceived physical and mental vigor, attenuate feelings of fatigue and low energy [39]. The strengthening of abdominal and back muscles could minimize the risk of lower back pain [34]. Noticeably, a quick rise in blood pressure and intra-abdominal pressure is triggered by heavy strength training during pregnancy, which may momentarily reduce blood flow to the fetus. However, resistance exercise with a modest to moderate load had no negative consequences during pregnancy [40]. Resistance exercise with elastic belts for 2–3 repetitive sets of movements, or through self-muscle exercises that involve upper and lower limbs for 2–3 sets of 12–24 repetitions, would fit well in pregnancy [41]. HIIT is the second most popular kind of exercise in the European fitness trends [42]. The intensity of training and interval section was monitored and measured by maximal heart rate or Borg’s rate of perceived exertion scale [43], meanwhile the ratio of exercise and rest time depends on individual capabilities, training progression, and pregnancy stage [44]. With regards to obstetric outcomes, HIIT programs were proven to be safe and well-accepted by pregnant women regardless of the training components and interval structure [45], even during the third trimester [46]. Nonetheless, the association between health outcomes and different type, timing, or domain of activity patterns cannot be assured due to insufficient research evidence [33].

Generally, PA during pregnancy with moderate intensity is agreed to be basically safe and beneficial for both mother and fetus [47,48,49]. However, exercise with contraindications is outside the purview of what an exercise expert is allowed to do. As official guidelines stated, women with absolute contraindications should be restricted from strenuous exercise. Women with relative contraindications should only participate in moderate-to-vigorous intensity PA with professional advice and obstetric care [37,50,51]. Gynecologists or other obstetric care professionals should thoroughly assess women with medical or obstetric complications before they begin the activities. Both active and inactive healthy women should start prenatal activities [34]. Women who suffered from complicated pregnancies (Gestational diabetes mellitus, high blood pressure, or other complications) also should continue their normal everyday activities and modify their exercise programs with qualified prenatal exercise specialists.

Given that “lifestyle intervention,” “leisure-time PA,” “sedentary behavior,” and “lifestyle” are all strong burst keywords, “sitting time” is one main cluster, and the effect of lifestyle intervention on pregnancy is one of the important research hotspots. Furthermore, “sedentary behavior” and “lifestyle” are also the newest strong burst keywords, which leads the effect of lifestyle intervention on pregnancy to be the research frontier on PA during pregnancy. Previous studies suggested that sedentary behaviour in pregnancy would probably lead to macrosomic infants, gestational weight gain, and hypertensive disorders [52], shorter gestation, and inhibited fetal growth [53]. That changing lifestyle during pregnancy by implementing leisure-time PA has been analyzed and discussed frequently in recent years [54,55,56]. A systematic review clearly showed that healthy expectant mothers can perform mild, moderate, and even vigorous levels of leisure-time PA without running the danger of giving birth prematurely [57]. A great number of previous cohort studies focused on diverse aspects can be found published, which is one of the foremost reasons why the #8 cohort study is presented as one main cluster. In maternal health, a systematic review that included seven cohort studies achieved controversial results, but generally, high-intensity leisure-time PA before and/or during pregnancy, or performing more than 4 h leisure-time PA each week may reduce the risk of pre-eclampsia [56]. A cohort study with 3209 participants suggested that leisure-time PA during both pre-pregnancy and early pregnancy reduced 46% of the risk of GDM compared with inactivity groups during both periods [58]. Meanwhile, another cohort study with 10,038 pregnant women illustrated that sustained low leisure-time PA during pregnancy is associated with excess risk of GDM and overall preterm birth. Moreover, women who increased leisure time PA lowered the rate of GDM. This could indicate that the increase in leisure-time PA in early pregnancy phases may be related to improved pregnancy health [59]. In offspring health, a cohort study proved that sons of women with light and moderate to heavy leisure-time PA had a lower risk of having a low intelligence score compared with sons of sedentary women [60]. Similarly, a systematic review and meta-analysis of 30 randomized controlled trials and 51 cohort studies supported the promotion of LTPA in pregnancy as a strategy to improve maternal and child health [61]. Furthermore, current cohort study results have also shown positive relationships between leisure-time PA and hyperemesis gravidarum [62], head circumference among male infants [63], birthweight among female infants, and women with normal prepregnancy BMI [63]. Conversely, no association was found between leisure-time PA and adiposity in mid-childhood [64] and intelligence quotient [65].

To our understanding, this is the first bibliometric study on PA during pregnancy. Due to the format specifications of the CiteSpace software, only papers from the Web of Science Core Collection database were retrieved. All the information was retrieved on 4 July 2022, research published after this date was not included and analyzed. Additionally, the majority of publications included were in English; thus, even though our retrieval approach did not restrict language, there might have been a linguistic bias.

## 5. Conclusions

The number of papers published on the topic of PA during pregnancy is increasing year by year. Research disciplines are accompanied by crossover and integration. The United States has made the most significant contribution to the research on this topic, ranking first in the world in both the number and centrality of research. The authors and institutions of the research published have more inter-country collaborations. Different patterns of PA, prevention, and amelioration of pregnancy complications are major research hotspots. Improvement of sedentary behaviour, lifestyle intervention through leisure-time PA, and preterm care are major research frontiers and have received extensive attention in recent years. Therefore, further research on evidence-based recommendations on popular PA during pregnancy, such as HIIT, should be globally implemented and promoted.

## Figures and Tables

**Figure 1 ijerph-19-14516-f001:**
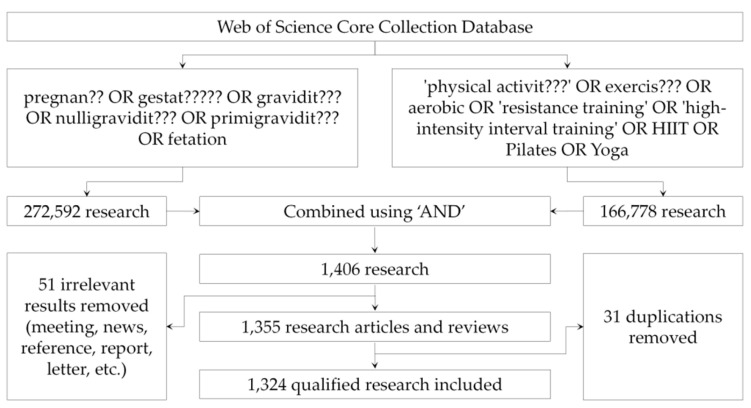
Retrieval strategy.

**Figure 2 ijerph-19-14516-f002:**
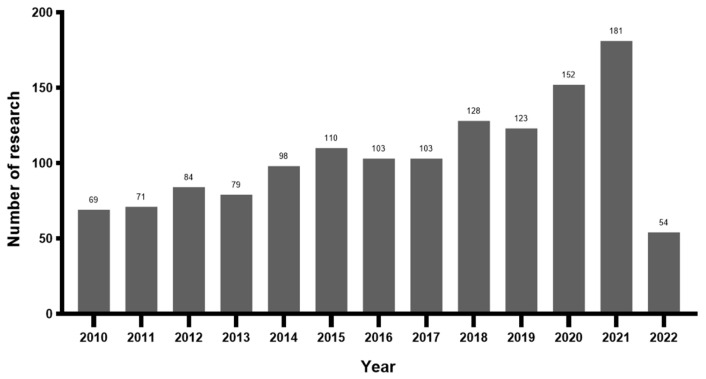
Distribution of publications by year in research on physical activity during pregnancy.

**Figure 3 ijerph-19-14516-f003:**
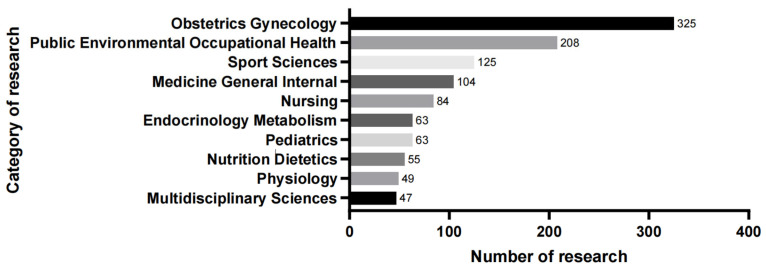
Distribution of categories in research of physical activity during pregnancy.

**Figure 4 ijerph-19-14516-f004:**
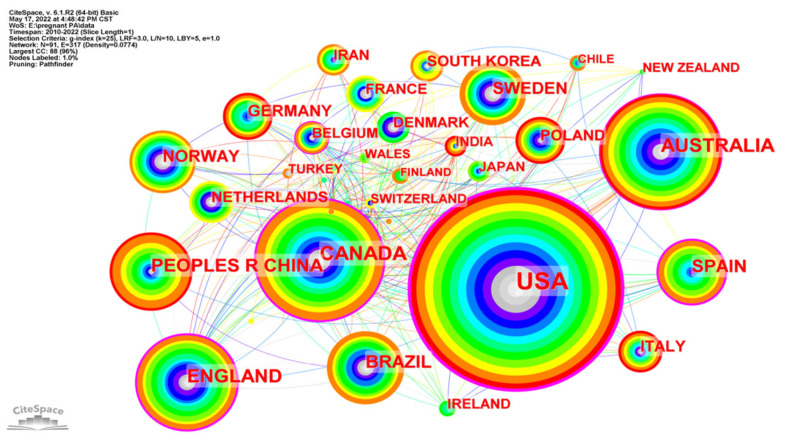
Co-occurrence map of countries researching physical activity during pregnancy.

**Figure 5 ijerph-19-14516-f005:**
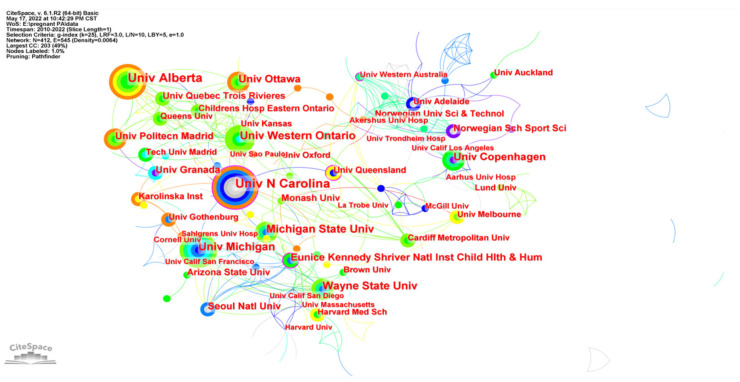
Co-occurrence map of institutions researching physical activity during pregnancy.

**Figure 6 ijerph-19-14516-f006:**
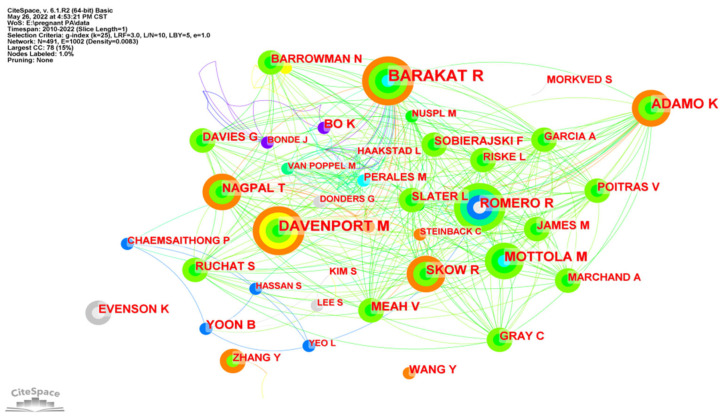
Co-occurrence map of authors researching physical activity during pregnancy.

**Figure 7 ijerph-19-14516-f007:**
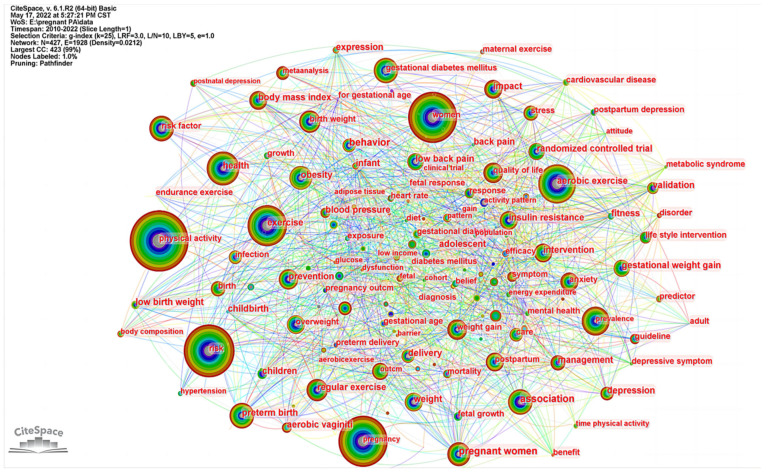
Co-occurrence map of keywords in research of physical activity during pregnancy.

**Figure 8 ijerph-19-14516-f008:**
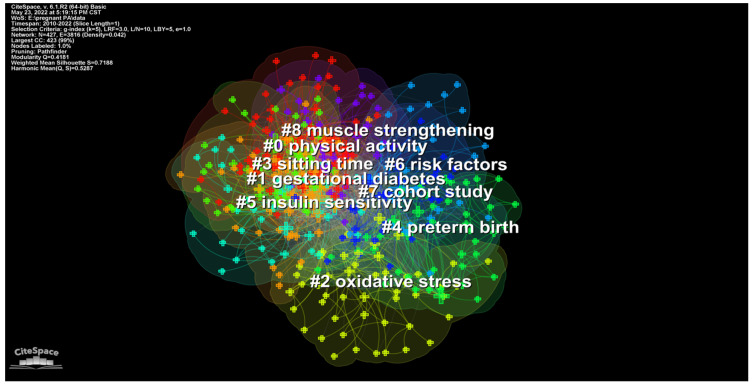
Cluster map of keywords in research on physical activity during pregnancy.

**Figure 9 ijerph-19-14516-f009:**
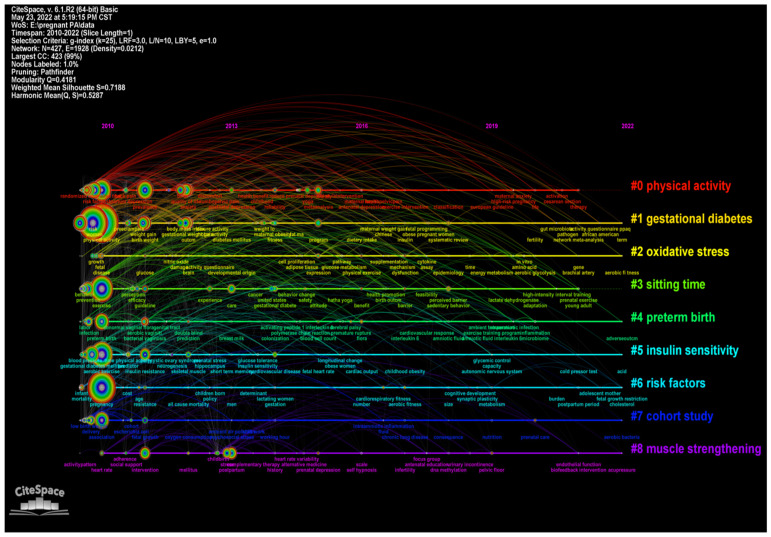
Timeline map of keywords in research of physical activity during pregnancy.

**Table 1 ijerph-19-14516-t001:** Top 10 productive countries/institutions in research of physical activity during pregnancy.

Rank	Country	Research Volume (Percentage)	Centrality	Rank	Institution	Research Volume (Percentage)	Centrality
1	USA	402 (35.5%)	0.55	1	University of Alberta	30 (14.6%)	0.01
2	Canada	131 (11.6%)	0.21	2	University of Carolina	29 (14.1%)	0.18
3	Australia	113 (10.0%)	0.17	3	University of Western Ontario	21 (10.2%)	0.06
4	England	100 (8.8%)	0.29	4	Michigan State University	21 (10.2%)	0.03
5	China	83 (7.3%)	0.01	5	University of Michigan	21 (10.2%)	0.03
6	Brazil	77 (6.8%)	0.03	6	Wayne State University	19 (9.2%)	0.01
7	Spain	74 (6.5%)	0.22	7	University of Copenhagen	18 (8.7%)	0.07
8	Norway	54 (4.8%)	0.04	8	University of Ottawa	17 (8.3%)	0
9	Sweden	53 (4.7%)	0.06	9	Eunice Kennedy Shriver Natl Inst Child Hlth & Hum	16 (7.8%)	0.12
10	Germany	45 (4.0%)	0.03	10	Seoul Natl University	14 (6.8%)	0.01

**Table 2 ijerph-19-14516-t002:** Top 10 productive authors/journals in research of physical activity during pregnancy.

Rank	Author	Country	Organization	Volume	Rank	Journal	5-IF	JCR	Volume (Percentage)
1	Barakat Ruben	USA	Northwell Health’s Cancer Institute	28	1	American Journal of Obstetrics and Gynecology	8.661	Q1	637 (14.8%)
2	Margie Davenport	Canada	University of Alberta	21	2	Obstetrics andGynecology	7.661	Q1	631 (14.7%)
3	Kristi Adamo	Canada	University of Ottawa	17	3	Medicine and Science in Sports and Exercise	5.441	Q1	509 (11.9%)
4	Michelle F. Mottola	Canada	University of Western Ontario	16	4	PLoS One	3.240	Q2	433 (10.1%)
5	Roberto Romero	USA	National Institutes of Health	14	5	Acta Obstetricia Et Gynecologica Scandinavica	3.636	Q2	370 (8.6%)
6	Rachel Skow	USA	University of Texas at Arlington	13	6	The Lancet	79.321	Q1	358 (8.3%)
7	Linda May	USA	East Carolina University	12	7	British Journal of Obstetrics and Gynaecology	7.661	Q1	356 (8.3%)
8	Victoria Meah	Canada	University of Alberta	12	8	BMC Pregnancy and Childbirth	3.007	Q2	355 (8.3%)
9	Taniya Nagpal	Canada	University of Western Ontario	12	9	Cochrane Database of Systematic Reviews	9.266	Q1	326 (7.6%)
10	Kari Bø	Norway	Akershus University Hospital	11	10	British Journal of Sports Medicine	13.800	Q1	317 (7.4%)

**Table 3 ijerph-19-14516-t003:** Top 20 keywords with the strongest citation bursts.

Rank	Keywords	Strength	Begin	End	2010–2022
1	activity pattern	7.35	2010	2015	▃▃▃▃▃▃▂▂▂▂▂▂▂
2	oxidative stress	4.43	2010	2012	▃▃▃▂▂▂▂▂▂▂▂▂▂
3	exercise	3.83	2010	2011	▃▃▂▂▂▂▂▂▂▂▂▂▂
4	weight	3.62	2010	2014	▃▃▃▃▃▂▂▂▂▂▂▂▂
5	maternal exercise	3.51	2010	2012	▃▃▃▂▂▂▂▂▂▂▂▂▂
6	endurance exercise	3.35	2010	2012	▃▃▃▂▂▂▂▂▂▂▂▂▂
7	leisure-time physical activity	4.45	2011	2012	▂▃▃▂▂▂▂▂▂▂▂▂▂
8	disability	4.01	2012	2014	▂▂▃▃▃▂▂▂▂▂▂▂▂
9	neurogenesis	3.46	2012	2013	▂▂▃▃▃▂▂▂▂▂▂▂▂
10	randomized controlled trial	3.32	2016	2018	▂▂▂▂▂▃▃▃▃▂▂▂▂
11	lifestyle intervention	4.53	2017	2019	▂▂▂▂▂▂▂▃▃▃▂▂▂
12	maternal obesity	3.57	2017	2018	▂▂▂▂▂▂▂▃▃▂▂▂▂
13	sedentary behavior	3.73	2018	2022	▂▂▂▂▂▂▂▂▃▃▃▃▃
14	systematic review	3.4	2018	2022	▂▂▂▂▂▂▂▂▃▃▃▃▃
15	birth	3.35	2018	2019	▂▂▂▂▂▂▂▂▃▃▂▂▂
16	weight loss	3.34	2018	2019	▂▂▂▂▂▂▂▂▃▃▂▂▂
17	symptom	7.42	2019	2022	▂▂▂▂▂▂▂▂▂▃▃▃▃
18	gestational diabetes mellitus	4.19	2019	2020	▂▂▂▂▂▂▂▂▂▃▃▂▂
19	lifestyle	3.42	2020	2022	▂▂▂▂▂▂▂▂▂▂▃▃▃
20	prenatal care	3.27	2020	2022	▂▂▂▂▂▂▂▂▂▂▃▃▃

## Data Availability

Not applicable.

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
