# Peer review of "Current Research Trends, Hotspots, and Frontiers of Physical Activity during Pregnancy: A Bibliometric Analysis"

_ijerph, 2022, doi:10.3390/ijerph192114516_

Round 1

Reviewer 1 Report

Review Table 1. retrieval strategy. Question marks and misprints for keywords. Consider the possibility of including a flow chart. Figure 6 and 8 are not seen correctly, it would be appropriate to improve it.

Reviewer 2 Report

I suggest that the 2018 Physical Activity Guidelines Advisory Committee Scientific Report be included as a featured document. PART F. CHAPTER 8. WOMEN WHO ARE PREGNANT OR POSTPARTUM

I suggest further discussion of the state of the art regarding type of physical activity, weekly frequency, specific exercises, and contraindications.

Reviewer 3 Report

I’d like to thank the authors and the Editorial Board for the opportunity to review the paper submitted to the International Journal of Environmental Research and Public Health. The manuscript entitled " Current research trends, hotspots and frontiers of physical activity during pregnancy: a bibliometric analysis ‘’ needs to be improved. I have the following comments to improve the manuscript: 1. Figures are illegible, difficult to interpret, please consider presenting them in a different graphic form 2. In the discussion I propose to move lines 155-167 and place them in the introduction to the article 3. References written not in accordance with the requirements set by the journal. A large part of the literature is older than the last 5 years. 4. The main perpose of the work is missing in the article. 5. The article does not add any new content

Round 2

Reviewer 3 Report

Tkanks  to the authors for making corrections to the manuscript